# VAM: Value-Attention Merging for KV Cache Optimization in LLMs

## Abstract

Efficient key-value (KV) cache management is essential for large language models (LLMs) performing long-text inference. Traditional methods, which retain all original KV pairs, lead to high memory usage and degraded performance due to outdated contextual representations. While existing solutions predominantly focus on cache eviction or compression to reduce memory and computation, they largely neglect the issue of semantic degradation in the cache itself. In this paper, we identify two critical limitations in long-context inference—Progressive Clustering and Context Degradation—which cause the model to lose global contextual awareness over time. To address these issues, we propose VAM, a plug-and-play KV cache optimization algorithm that dynamically merges attention outputs into value states. Unlike cache compression methods that aim to reduce cache size, VAM specifically targets the preservation of contextual semantics in the cached representations, thereby improving the model's ability to retain and utilize long-range dependencies. VAM is lightweight, easy to integrate, and complementary to existing compression strategies. Experiments on LongBench tasks across LLaMA and Mistral models (7B–70B) show consistent improvements of 0.36–6.45 in absolute score (0.64%–4.26% relative), and up to 8.33% when combined with state-of-the-art KV compression methods, demonstrating VAM's effectiveness in enhancing long-sequence inference quality. Our code is available at https://anonymous.4open.science/r/vam-torch-386B/.

## 1 Introduction

Long-text inference plays a pivotal role in enabling large language models (LLMs) to excel in applications such as book summarization, legal document analysis, and multi-turn dialogues Li et al. (2024a); Zhang et al. (2020); Chalkidis et al. (2020); Song et al. (2022). By effectively processing extended contexts, LLMs generate coherent and contextually rich output, addressing tasks that require a comprehensive understanding of long or complex information. As models like GPT-4 Achiam et al. (2023) and Gemini 1.5 Pro Team et al. (2024) expand their context-length capabilities to hundreds of thousands or even millions of tokens, they unlock unprecedented potential for real-world scenarios that rely on long-text reasoning Liu et al. (2024a).

The KV cache Pope et al. (2023) is a common optimization in LLM inference, reducing the attention mechanism's complexity from quadratic to linear by storing the key and value tensors of previous tokens. However, in long-text inference, the cache size grows linearly with sequence length, often exceeding the memory required for model weights. This leads to significant GPU memory overhead, complicating deployment in resource-constrained environments. To address this, methods such as token eviction Zhang et al. (2023); Liu et al. (2024b), KV cache quantization Hooper et al. (2024); Liu et al. (2024c), and other compression techniques Yang et al. (2024); Sun et al. (2024); Wan et al. (2024) focus on reducing the memory footprint of the KV cache.

In addition to the well-known challenge of high memory consumption, we identify a critical yet underexplored limitation of the KV cache: its inability to capture higher-level semantic and structural relationships, such as discourse dependencies and topic continuity, which are essential for tasks like summarization, story generation, and multi-turn dialogue. To better understand this issue, we analyze t-SNE visualizations of LLaMA2-7B inference and uncover two key phenomena: *Progressive Clustering*, where token embeddings become increasingly localized as the sequence lengthens, and

*Context Degradation*, where the model increasingly prioritizes recent tokens, weakening its ability to maintain long-range dependencies. These phenomena, shaped by the cumulative effect of Softmax over time, reveal a key gap in existing research that this paper addresses.

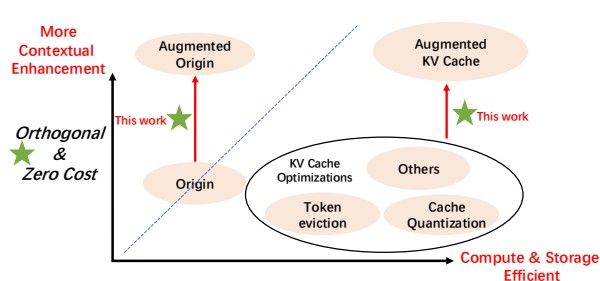

Figure 1: Design space of KV cache optimizations. Existing methods (e.g., token eviction, quantization) focus on efficiency, while ours enhances contextual representation with zero overhead. VAM integrates seamlessly with Origin and other optimizations, demonstrating orthogonality and complementarity.

In this paper, we propose a technique, Value-Attention Merging (VAM), that enhances the KV cache in LLMs by fusing the value tensor with the attention output and addressing its limitation of overlooking contextual information. This fusion method preserves both token-level and higher-order contextual insights, improving the capture of long-range dependencies and semantic relationships. As a result, it enhances performance on tasks like document-level reasoning and multi-turn dialogue, while reducing redundant attention computations. This balance between efficient storage and contextual expressiveness enables scalable and accurate long-text processing.

As a plug-and-play approach, VAM requires no additional training, introduces minimal modifications to the inference process, and delivers measurable performance improvements with negligible time and space overhead. Specifically designed to tackle long-text tasks, this lightweight and effective method enhances the model's ability to maintain long-range dependencies. Furthermore, its orthogonality to other KV cache optimization techniques increases its applicability, making it a versatile solution for improving LLM performance on long-text inference tasks. Figure 1 illustrates the design space for KV cache optimizations.

The contributions of this work are summarized as follows:

- **Identification of Key Limitations in KV Cache:** We highlight the challenges of *Progressive Clustering* and *Context Degradation* in traditional KV cache mechanisms, which impair long-range semantic retention essential for tasks like discourse and topic continuity.
- **VAM for KV Cache Optimization**: We propose VAM, a simple yet effective optimization technique that fuses value tensors with attention outputs to better capture long-range dependencies, thereby improving contextual understanding without additional computational cost.
- **Comprehensive Empirical Validation**: Extensive experiments on LLaMA and Mistral models show that our method consistently improves LongBench performance, with absolute gains of 0.36 to 6.45 and relative improvements of 0.64% to 4.26%, demonstrating robust long-text understanding. When used as a plug-in to existing KV cache optimization methods, it further enhances performance by up to 8.33%, validating its effectiveness across architectures.

## 2 MOTIVATION

This section investigates context degradation during long-sequence inference with traditional KV caching in LLMs. Using t-SNE visualizations, we analyze token distribution evolution throughout inference, revealing a shift from global to local attention that motivates our novel strategy.

### 2.1 PROBLEM INTRODUCTION

The KV cache mechanism stores intermediate states (keys and values) for efficient self-attention calculations in LLMs. However, during long-sequence inference, the cache grows proportionally with input length, creating significant computational and memory overhead. This expansion strains hardware resources and challenges real-time inference for tasks involving extensive contexts.

The increase in the size of the KV cache does not always translate to improved model performance. During inference, the attention mechanism initially captures dependencies across the entire sequence, operating in a global context. However, as the sequence length grows, the attention weights begin to prioritize more recent tokens, diminishing the contribution of earlier tokens. This shift from a global to a local focus reduces the KV cache's ability to maintain long-term dependencies, leading to performance inefficiencies despite the growing storage demands.

These challenges highlight the need for efficient KV cache management that minimizes storage while preserving the model's ability to capture both global and local contexts.

## 2.2 T-SNE ANALYSIS OF CONTEXT DEGRADATION

To understand KV cache behavior during long-sequence inference, we visualize token embedding evolution using t-SNE. We record value states from LLaMA2-7B's KV cache and project these high-dimensional representations into 2D space, revealing evolving attention patterns (Figure 2).

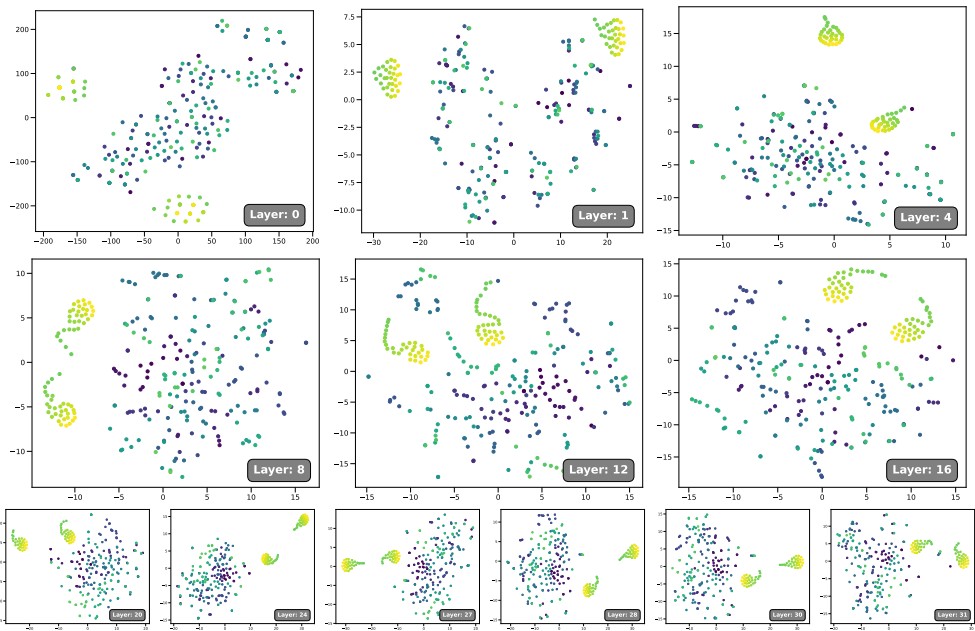

Figure 2: t-SNE visualization of the value vectors stored in the KV cache of the LLaMA2-7B model across different layers. Token positions are color-coded, with darker colors indicating earlier positions in the sequence and lighter colors representing later positions. As the token position increases, the token distributions transition from a globally dispersed pattern to increasingly localized clusters dominated by lighter-colored tokens. This progressive clustering of later-position tokens highlights the context degradation phenomenon, where the model increasingly attends to recent tokens while earlier ones lose influence during inference.

The t-SNE visualization of KV cache values reveals how token distributions evolve during inference, exposing attention mechanism dynamics and limitations in handling long sequences. Three key phenomena emerge:

**Attention Equilibrium.** Early in inference, token embeddings are widely distributed in 2D space (dark points in Figure 2), reflecting the model's global dependency capture. Tokens from the entire sequence are equally represented in the KV cache, with the diverse spread indicating attention to broad contextual elements for coherent predictions. At this stage, the KV cache operates optimally, with all tokens contributing meaningfully to attention.

**Progressive Clustering.** As inference progresses, token embeddings cluster into localized regions (light-colored points in Figure 2), reflecting increased focus on recent tokens while de-prioritizing earlier ones. Cluster centers shift away from earlier contexts, indicating reduced global coverage. Softmax-driven attention weight decay for distant tokens enhances short-term efficiency but dimin-

ishes long-range dependency retention. This marks context degradation onset as attention range narrows.

**Context Degradation.** In later inference stages, token embeddings form compact clusters, showing intensified focus on recent context while de-prioritizing earlier tokens. This sharpens short-term dependency handling but weakens long-term retention. Earlier tokens remain in the KV cache but contribute minimally to predictions, creating inefficiency where maintenance costs exceed utility.

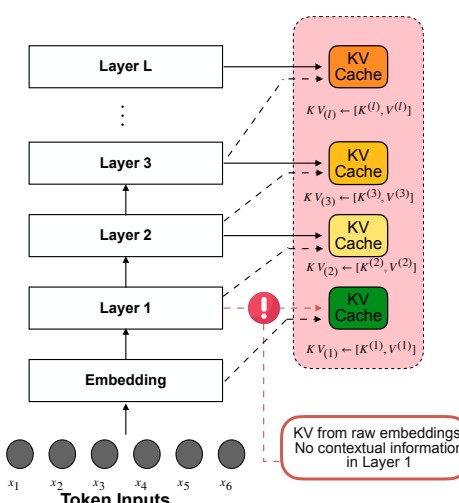

Figure 3: The first-layer KV cache lacks context, leading to misaligned attention and degraded long-range dependency modeling.

### 2.3 ANALYSIS

In multi-layer LLMs, the KV cache updates before each attention computation. At the first layer, the cache contains only raw token embeddings without contextual information, as no attention output has been computed yet. This initial context absence creates errors that propagate through subsequent layers—each layer operates on increasingly flawed inputs from previous layers (Figure 3). The misalignment widens gaps between contextually related tokens, impairing long-range dependency tracking. Consequently, the attention mechanism shifts toward localized, short-term dependencies, degrading long-range attention capabilities.

To address this issue, we must transform the KV cache from a static storage mechanism into a dynamic, adaptive system that continuously incorporates contextual information. The key insight is injecting the first layer's attention output directly into the KV cache, enabling the model to capture contextual relationships from the initial computation and preventing propagation of incomplete token representations through subsequent layers. This dynamic update mechanism resolves the initial context absence while preserving representation integrity throughout inference.

## 3 RELATED WORK AND SHORTCOMINGS

This section reviews existing KV cache optimization approaches for transformer-based LLMs. While these methods address memory and computational overhead, they fail to resolve context degradation and stale representations (Section 2). We examine their limitations to establish the foundation for our proposed approach.

**Cache Compression.** There has been substantial prior work on compressing the KV cache, focusing on retaining important tokens while evicting less significant ones to reduce memory usage Ge et al. (2023); Adnan et al. (2024). Some methods also retrieve only a subset of tokens at each step to save memory bandwidth Ribar et al. (2023). Recent approaches, such as H2O Zhang et al. (2023), PyramidKV Cai et al. (2024), and SnapKV Li et al. (2024b), optimize memory efficiency by selecting high-utility tokens based on attention scores or token utility patterns. While these methods improve memory usage and decoding speed, they often rely on heuristics or fixed observation windows, which can struggle with preserving context in long sequences. More recent efforts explore norm-based metrics, such as using the value vector norm Guo et al. (2024) or key vector norm Devoto et al. (2024) to estimate token utility, providing a simple yet effective alternative to attention-based heuristics.

**Quantization.** Extensive research on LLM quantization aims to reduce memory and runtime costs. Weight quantization has led to techniques like dense-and-sparse decomposition Dettmers et al. (2023); Kim et al. (2023) and non-uniform schemes using adaptive clustering Dettmers et al. (2024). For KV cache, methods such as ZipCache He et al. (2024) and GEAR Kang et al. (2024) achieve efficient compression via per-channel quantization and low-rank representations, though preserving accuracy over long sequences remains challenging. Other approaches extend quantization to activations and

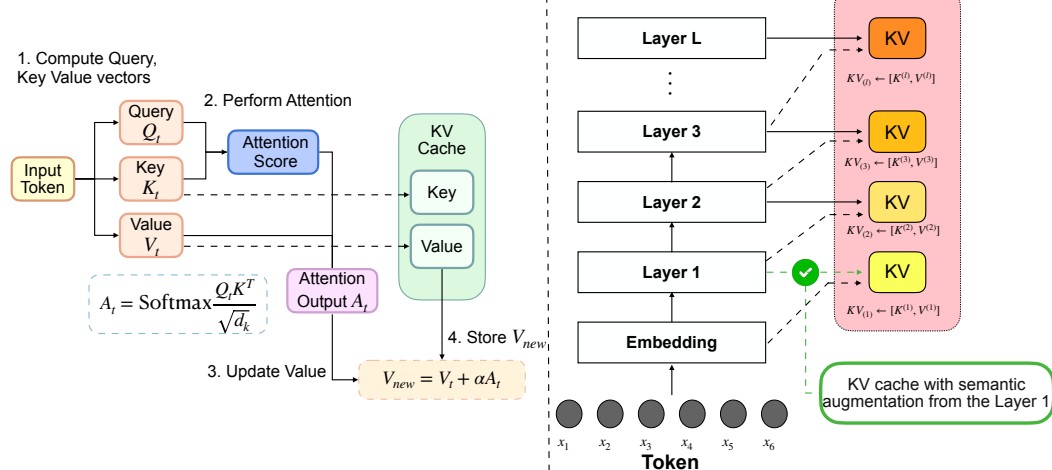

Figure 4: Illustration of the VAMP algorithm for optimized KV cache update, where value are updated with a weighted sum of attention using a factor $\alpha$. This method enhances inference efficiency by minimizing redundant computations and provides a simple, plug-and-play design for seamless integration with existing systems.

KV cache jointly, reaching 4-bit precision with fine-grained grouping, albeit with perplexity trade-offs Sheng et al. (2023); Zhao et al. (2024).

Existing KV cache optimization methods are often complex and difficult to implement. These approaches involve advanced techniques like non-uniform quantization, matrix approximations, and token selection strategies, all requiring careful optimization and fine-tuning. Additionally, some methods necessitate custom hardware or specialized software for full effectiveness. The complexity and resource demands of these methods make them challenging to scale, highlighting the need for simpler, more accessible solutions.

## 4 VALUE-ATTENTION MERGING

In this section, we introduce VAM, a novel key-value cache optimization framework designed to mitigate context degradation and enhance efficiency in Transformer-based models. We begin with an overview of the methodology (Section 4.1), followed by a detailed explanation of its mathematical foundation (Section 4.2), and conclude with practical implementation details (Section 4.3).

### 4.1 OVERVIEW

Figure 4 illustrates the architecture of VAM, our proposed KV cache optimization framework. VAM dynamically updates the KV cache by merging static value representations with attention output. The core design principle of VAM is to integrate dynamic contextual information into the KV cache, thereby mitigating context degradation and preventing stale representations. Unlike traditional KV cache mechanisms that store static key-value pairs, VAM continuously refreshes the value representations to ensure their relevance throughout the inference process. This dynamic updating mechanism enhances the model's ability to maintain coherent and contextually accurate representations, especially during extended autoregressive generation tasks.

### 4.2 MATHEMATICAL FOUNDATION

Transformer models use the attention mechanism to compute contextual representations for each token by attending to all tokens within the sequence. This mechanism relies on three primary components: the query ($Q$), the key ($K$), and the value ($V$), all derived from the input embeddings. The attention

operation is mathematically defined as:

$$\text{Attention}(Q, K, V) = \text{Softmax}\left(\frac{QK^\top}{\sqrt{d_k}}\right) V. \tag{1}$$

Here, $d_k$ denotes the dimensionality of the key vectors, and the softmax function ensures that the attention weights sum to one, facilitating a weighted sum of the value vectors $V$.

During inference, particularly in autoregressive generation, the key-value cache stores the key and value matrices to prevent redundant computations for previously processed tokens, thereby improving efficiency. The cache update mechanism is formalized as follows.

$$K_{\text{cache}} = [K_1; K_2; \ldots; K_t], V_{\text{cache}} = [V_1; V_2; \ldots; V_t], \tag{2}$$

where $K_t$ and $V_t$ denote the key and value matrices for the token at timestep $t$, respectively. At each inference step, the attention mechanism uses the cached keys and values, along with the current query $Q_t$ to compute the attention output:

$$\text{Attention}(Q_t, K_{\text{cache}}, V_{\text{cache}}) = \text{softmax}\left(\frac{Q_t K_{\text{cache}}^\top}{\sqrt{d_k}}\right) V_{\text{cache}}. \tag{3}$$

This formulation ensures that each new token integrates information from all preceding tokens without necessitating the recomputation of their representations, thereby optimizing the inference process.

### 4.3 Dynamic KV Cache Optimization with VAM

VAM introduces a dynamic mechanism for updating the KV cache during autoregressive inference in Transformer models. Its primary goal is to address the limitations of static KV caches, such as stale representations and context degradation, while ensuring seamless compatibility with existing Transformer architectures.

VAM fundamentally alters the standard KV cache update procedure by dynamically integrating the output of the attention mechanism directly into the value vectors. In traditional Transformer-based models, the KV cache is updated by sequentially appending static key-value pairs as tokens are processed. Specifically, at each timestep $t$, the key vectors $K_t$ and value vector $V_t$ are computed and appended to their respective caches without modification:

$$K_{\text{cache}}^{(t)} = \left[K_{\text{cache}}^{(t-1)}, K_t\right], \quad V_{\text{cache}}^{(t)} = \left[V_{\text{cache}}^{(t-1)}, V_t\right]. \tag{4}$$

While this static approach is efficient, it can lead to context degradation and the accumulation of stale representations, particularly in long sequences where the relevance of earlier tokens diminishes over time.

VAM addresses these limitations by modifying the update mechanism for the value cache. Rather than directly appending the static value vectors $V_t$, VAM dynamically updates $V_t$ by integrating them with the attention output. This integration is controlled by a hyperparameter $\alpha$, which determines the degree to which the attention output influences the updated value vectors:

$$V_t^{\text{updated}} = V_t + \alpha \cdot A_t. \tag{5}$$

The updated value vector $V_t^{updated}$ retains token-specific information from $V_t$ while integrating contextual insights from the attention output $A_t$. This approach resolves context absence in the first layer and mitigates context degradation, ensuring that cached representations remain relevant and up-to-date, thus preserving the integrity of the generated outputs over long sequences.

The updated value vector is then appended to the value cache:

$$V_{\text{cache}}^{(t)} = \left[V_{\text{cache}}^{(t-1)}; V_t^{\text{updated}}\right]. \tag{6}$$

VAM enhances the Transformer's value cache by dynamically merging each value vector with the corresponding attention output, controlled by a single hyperparameter $\alpha$. A coarse grid search on LongBench Bai et al. (2024) tasks shows that $\alpha \in [0.3, 0.7]$ achieves the best trade-off—smaller values fail to inject sufficient context, while larger values risk overwriting token semantics. This range consistently delivers stable gains and strong overall performance across models and tasks,

offering a reliable balance between contextual expressiveness and semantic fidelity without the need for task-specific tuning.

VAM modifies only the value cache, preserving full compatibility with existing Transformer architectures and acceleration frameworks like FlashAttention Dao et al. (2022) and Group-Query Attention Ainslie et al. (2023). It requires no change to the attention computation and can be seamlessly integrated into existing inference pipelines.

VAM brings two key benefits: (1) it improves long-range context retention by preventing stale value accumulation; and (2) it accelerates semantic flow from the first layer, addressing the cold-start issue. These advantages enhance generation quality with negligible overhead.

## 5 EXPERIMENTS

In this section, we demonstrate that VAM, a simple yet effective KV cache optimization strategy, significantly enhances long-text inference by improving memory efficiency, reducing latency, and maintaining or even improving model performance. With minimal computational overhead, VAM seamlessly integrates with existing KV cache mechanisms, making it highly applicable to various LLMs and real-world tasks.

### 5.1 EXPERIMENTAL SETTINGS

**Models.** We conducted experiments on a diverse set of models, including the LLaMA and Mistral series, with parameter sizes ranging from 7B to 70B. These models were selected for their widespread adoption and architectural diversity, allowing for a comprehensive evaluation of our method across varying scales and design paradigms.

**Evaluation Strategies.** (1) Long-sequence Language Modeling: We first evaluate the performance of our proposed method by comparing it with the original inference approach using LongBench Bai et al. (2024), a comprehensive benchmarking suite designed to assess language models across a range of tasks. (2) Plug-in Effectiveness. We evaluate VAM as a plug-in enhancement to existing KV cache compression methods, including H2O Zhang et al. (2023), PyramidKV Cai et al. (2024), SnapKV Li et al. (2024b), and ZipCache He et al. (2024). Experiments are conducted on Code, Few-shot learning, and Synthetic tasks, with detailed settings provided in Appendix B.(3) Retrieval-based Evaluation: Finally, we assess the precision of our method using the Needle-in-a-Haystack test, which tests the model's ability to retrieve specific information from large datasets.

### 5.2 MAIN RESULTS

**Long-sequence Language Modeling.** As shown in Table 1, our method achieves consistent and notable improvements across multiple LongBench tasks, with absolute gains ranging from 0.36 to 6.45 and relative improvements of 0.64% to 4.26% across different models. These results highlight the robustness of our approach in enhancing long-text understanding. The most stable gains are obtained with moderate $\alpha$ values (e.g., 0.1–1.0), which strike an effective balance between incorporating new contextual signals and preserving token-specific semantics. Although very large $\alpha$ values ($\alpha > 1.2$) may occasionally degrade performance, the overall trend clearly demonstrates that our method provides broad and reliable benefits across tasks and models.

**Enhancement of KV Cache Optimization.** In Table 2, we report the experimental results demonstrating that VAM consistently improves the performance of existing KV cache compression algorithms across diverse tasks from LongBench. With $\alpha = 0.35$, our method achieves substantial improvements on LLaMA3.1-8B-Instruct across all base methods and task categories, with improvements ranging from +0.01 (ZipCache on TriviaQA) to +3.00 (PyramidKV on TREC) across code understanding tasks (LCC and RepoBench-P), few-shot learning tasks (TREC and TriviaQA), and synthetic tasks (PassageCount). The average improvements per base method demonstrate consistent effectiveness: H2O (+1.04), PyramidKV (+1.17), SnapKV (+0.85), and ZipCache (+1.46), indicating that our method serves as a highly effective plug-in module that enhances the semantic capacity of compressed caches across different compression strategies.

Table 1: Comparison of performance across various LLMs under different configurations of the hyperparameter $\alpha$. For each model, the first (gray-shaded) row presents the baseline results obtained with the original KV cache without modification. Datasets are indexed by IDs, with their corresponding descriptions and evaluation metrics provided in Table 3 in Appendix C. **Bold** values highlight improvements over the baseline, while underlined values indicate the best performance within each setting. Our method outperforms the original approach in most cases (75%).

| Model | Setting($\alpha$) | Single-Doc QA | | | | Multi-Doc QA | | | | Summarization | | | | Avg. |
|---|---|---|---|---|---|---|---|---|---|---|---|---|---|---|
| | | 1-1 | 1-2 | 1-3 | 1-4 | 2-1 | 2-2 | 2-3 | 2-4 | 3-1 | 3-2 | 3-3 | 3-4 | |
| LLaMA-2-7B | - | 19.14 | 21.81 | 37.66 | 11.82 | 27.85 | 31.30 | 8.27 | 6.21 | 26.92 | 20.81 | 26.05 | 0.21 | *19.84* |
| | 0.1 | 19.24 | 22.31 | 36.82 | 12.06 | 27.91 | 31.57 | 8.21 | 6.44 | 27.29 | 20.96 | 26.24 | 0.21 | **19.94** |
| | 0.2 | 18.94 | 22.34 | 37.31 | 12.55 | 27.97 | 31.82 | 8.86 | 6.41 | 27.91 | 20.97 | 26.40 | 1.82 | **20.28** |
| | 0.35 | 19.00 | 22.23 | 36.54 | 12.41 | 27.91 | 31.71 | 8.53 | 7.14 | 27.48 | 20.91 | 26.19 | 2.17 | **20.18** |
| | 0.5 | 18.80 | 22.31 | 37.28 | 13.17 | 28.06 | 31.73 | 8.58 | 7.34 | 27.59 | 20.84 | 26.70 | 0.18 | **20.22** |
| | 1.0 | 19.23 | 20.78 | 36.14 | 12.87 | 27.27 | 31.91 | 8.54 | 5.09 | 23.18 | 19.84 | 24.71 | 0.36 | 19.16 |
| | 1.25 | 19.07 | 12.22 | 28.97 | 5.52 | 27.28 | 30.20 | 8.07 | 12.09 | 25.21 | 20.29 | 25.30 | 0.32 | 17.88 |
| Mistral-7B | - | 29.41 | 39.66 | 50.43 | 55.59 | 49.32 | 34.58 | 26.96 | 31.42 | 34.44 | 25.51 | 26.41 | 15.81 | *34.96* |
| | 0.1 | 29.42 | 39.72 | 50.46 | 55.66 | 49.29 | 34.79 | 26.97 | 32.69 | 34.36 | 25.73 | 26.70 | 16.08 | **35.15** |
| | 0.2 | 29.47 | 39.57 | 50.53 | 55.66 | 49.11 | 34.63 | 27.34 | 32.39 | 34.63 | 26.16 | 26.72 | 16.72 | **35.25** |
| | 0.35 | 29.06 | 39.98 | 50.69 | 55.88 | 49.37 | 34.22 | 27.22 | 31.63 | 34.96 | 25.86 | 27.38 | 17.22 | **35.29** |
| | 0.5 | 29.58 | 40.13 | 50.51 | 55.82 | 49.14 | 34.07 | 27.37 | 31.92 | 34.71 | 26.24 | 26.90 | 16.93 | **35.28** |
| | 1.0 | 28.62 | 40.27 | 49.97 | 55.96 | 50.03 | 34.09 | 27.84 | 31.91 | 34.89 | 25.36 | 26.13 | 17.06 | **35.18** |
| | 1.5 | 25.31 | 38.03 | 49.01 | 55.39 | 49.92 | 33.21 | 29.13 | 32.01 | 34.53 | 25.08 | 26.26 | 16.17 | 34.50 |
| LLama-3.1-8B | - | 24.23 | 21.84 | 40.21 | 37.56 | 42.06 | 33.86 | 21.01 | 30.66 | 34.91 | 24.27 | 27.33 | 17.17 | *29.59* |
| | 0.1 | 24.32 | 22.64 | 40.72 | 37.88 | 42.10 | 34.06 | 20.82 | 31.21 | 35.21 | 24.24 | 27.52 | 17.38 | **29.84** |
| | 0.2 | 24.18 | 22.96 | 40.74 | 37.85 | 42.31 | 34.00 | 21.37 | 31.16 | 35.26 | 24.17 | 27.57 | 17.72 | **29.94** |
| | 0.35 | 24.88 | 22.98 | 40.89 | 38.17 | 42.50 | 34.41 | 21.13 | 30.91 | 35.39 | 24.27 | 27.33 | 18.23 | **30.09** |
| | 0.5 | 24.90 | 22.53 | 41.00 | 37.83 | 42.31 | 34.08 | 21.55 | 31.55 | 36.01 | 23.63 | 27.45 | 17.34 | **30.01** |
| | 1.0 | 23.27 | 22.51 | 40.54 | 37.64 | 41.74 | 33.67 | 21.82 | 32.43 | 35.73 | 24.17 | 28.01 | 17.49 | **29.94** |
| | 1.5 | 24.33 | 22.30 | 40.03 | 36.66 | 41.38 | 33.67 | 20.88 | 32.20 | 18.62 | 24.96 | 21.04 | 16.79 | 27.74 |
| LLaMA-3.1-70B | - | 26.48 | 45.17 | 49.27 | 24.81 | 50.43 | 51.84 | 27.61 | 34.27 | 39.47 | 24.09 | 27.58 | 20.24 | *35.10* |
| | 0.1 | 26.54 | 45.33 | 49.36 | 25.08 | 51.27 | 51.96 | 27.27 | 34.48 | 40.37 | 24.11 | 27.54 | 20.29 | **35.30** |
| | 0.2 | 26.81 | 45.28 | 49.44 | 25.16 | 51.61 | 53.04 | 27.66 | 35.19 | 41.22 | 24.17 | 27.27 | 21.28 | **35.68** |
| | 0.35 | 26.87 | 45.62 | 49.72 | 24.75 | 50.74 | 52.77 | 28.28 | 34.72 | 41.08 | 25.22 | 27.62 | 21.44 | **35.65** |
| | 0.5 | 26.74 | 45.53 | 49.19 | 25.22 | 50.97 | 51.97 | 28.42 | 35.09 | 40.29 | 25.32 | 27.94 | 20.97 | **35.55** |
| | 1.0 | 25.92 | 45.17 | 49.13 | 24.93 | 50.24 | 51.83 | 27.08 | 34.18 | 40.07 | 23.87 | 26.87 | 21.06 | 35.03 |
| | 1.25 | 25.43 | 44.32 | 48.74 | 24.90 | 50.07 | 51.71 | 26.44 | 34.32 | 39.12 | 23.09 | 27.04 | 20.23 | 34.62 |

ᵃ Details of used LLMs in this table are presented in 4.

Table 2: Plug-in effectiveness of VAM: enhancing semantic capacity of compressed KV caches across different compression strategies on LongBench evaluation tasks using $\alpha = 0.35$.

| Base Method | LCC | RepoBench-P | TREC | TriviaQA | PssageCount | Avg. |
|---|---|---|---|---|---|---|
| **H2O** | 62.30 → 63.06 | 55.39 → **56.91** | 68.00 → **70.05** | 91.23 → **91.83** | 6.11 → **6.36** | **+1.04** |
| **PyramidKV** | 61.58 → **62.37** | 53.89 → **55.44** | 68.00 → **71.00** | 91.65 → **92.09** | 6.00 → **6.05** | **+1.17** |
| **SnapKV** | 62.62 → **63.23** | 56.56 → **56.66** | 68.00 → **70.50** | 91.48 → **92.03** | 6.00 → **6.50** | **+0.85** |
| **ZipCache** | 60.73 → **62.76** | 52.64 → **55.32** | 68.00 → **70.50** | 91.48 → **91.49** | 5.92 → **6.00** | **+1.46** |
| *Average improvement across all methods and datasets* | | | | | | **+1.13** |

Each cell displays baseline score → **VAM-enhanced score**. Underlined values indicate best performance per dataset.

**Retrieval-based Evaluation.** Figure 5 presents the experimental results comparing the original model (top) with our proposed method (bottom), where the $\alpha$ parameter in our method is set to 0.35. The white vertical line indicates the pre-trained window size. As observed, our method consistently achieves higher accuracy across various window sizes and insertion depths. The green regions, which represent higher accuracy, are more prevalent in our approach, particularly beyond the pre-trained window size. In contrast, the original model shows a noticeable decline in accuracy, especially as the window size increases and needle insertion depth deepens. These results demonstrate the effectiveness of our method in maintaining robust performance, even in more challenging scenarios, thereby improving the model's overall accuracy.

## 5.3 T-SNE VISUALIZATION: RESOLVING LONG-TEXT CONTEXTUAL DEGRADATION

In this experiment, we set $\alpha = 0.35$ and re-conducted t-SNE visualizations to assess the effect of our method on long-context inference. As shown in Figure 6 and Figure 7, our approach alleviates two major issues in the original model: the drift from global to local attention and the degradation

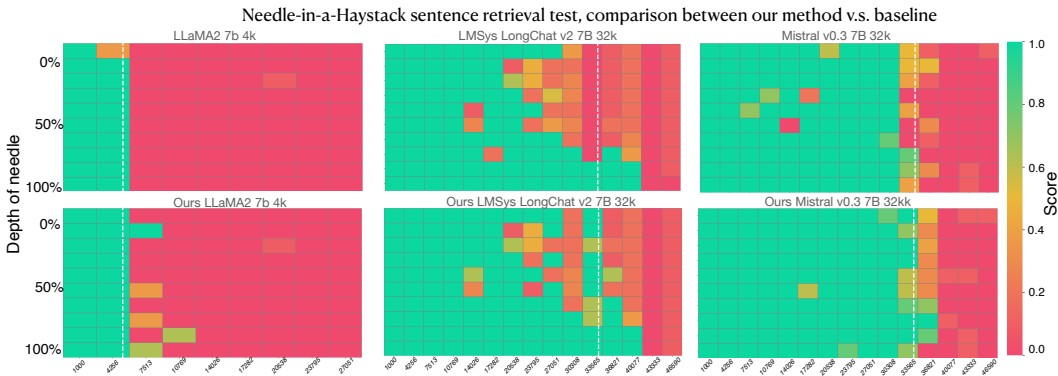

Figure 5: Results of the Needle-in-a-Haystack experiment comparing the original model (top) and our method (bottom). Green regions show higher accuracy, with our method excelling beyond the pre-trained window size (white line) and at deeper insertion depths.

of contextual representations. The resulting plots show more compact and uniform token clusters, indicating stronger global context retention. By augmenting cached values with attention outputs, our method mitigates delayed context propagation, enabling the model to preserve long-range dependencies and maintain coherence over extended sequences.

**(a) Baseline: Progressive Context Degradation Across Layers**

**(b) VAM: Consistent Semantic Structure Preservation**

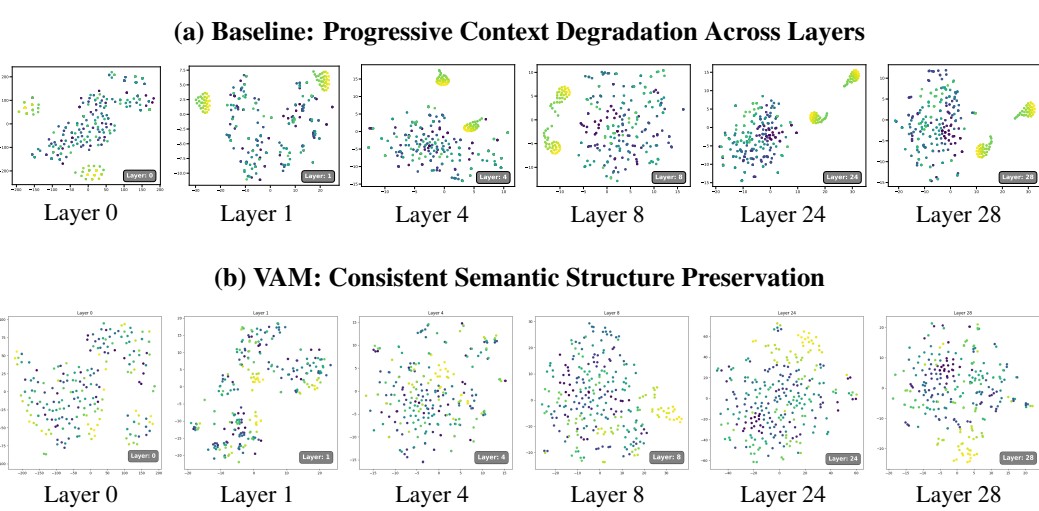

Figure 6: Layer-wise evolution of token representations in KV cache. (a) Baseline methods show progressive context degradation where early tokens (darker colors) become increasingly dispersed in deeper layers while recent tokens (lighter colors) dominate the representation space. (b) VAM maintains structured token distributions across all layers, ensuring balanced attention to both historical and recent context, effectively addressing the context degradation phenomenon.

## 6 CONCLUSION

In this paper, we introduced VAM to improve long-text inference in large language models by alleviating context degradation and the drift from global to local attention. Extensive experiments and t-SNE analyses demonstrate that VAM effectively preserves contextual coherence across long inference windows while maintaining computational efficiency. As a lightweight and easily integrable mechanism, VAM holds strong potential for real-world applications requiring long-context reasoning, and it also opens future directions such as combining with cache compression, retrieval-augmented inference, or multimodal extensions to further enhance scalability and robustness.

ETHICS STATEMENT

This work does not involve human subjects, sensitive personal data, or experiments that could directly cause harm. All datasets used are publicly available and widely adopted in the research community, and we followed their intended licensing and usage guidelines. Our methods focus on improving the efficiency and scalability of large language models without altering their underlying behaviors. We also considered potential risks such as model misuse, fairness, and bias, but our contributions are methodological and do not introduce additional ethical concerns. The authors have adhered to the ICLR Code of Ethics throughout the research and submission process.

REPRODUCIBILITY STATEMENT

We have taken multiple steps to ensure the reproducibility of our workps, are included in the appendix. Upon acceptance, we will release anonymized source code, scripts for running experiments, and dataset preprocessing instructions as supplementary materials. . Detailed descriptions of the model architectures, training configurations, and evaluation protocols are provided in the main text, while additional implementation details, such as hyperparameter settings, baseline configurations, and ablation study seturted in the paper can be regenerated using the released resources.

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

## A    ALGORITHMIC FLOW OF VAM

We present a detailed description of the algorithmic components of the proposed KV cache optimization framework, VAM. Unlike static KV caches, our framework dynamically updates the value vectors during autoregressive inference in Transformer models, thereby overcoming their inherent limitations. The step-by-step procedure is outlined below and illustrated in the accompanying flowchart, which shows how the KV cache is updated with attention outputs. The formal procedure is summarized in Algorithm 1.

## B    DETAILED EXPERIMENTAL SETUP

All experiments were conducted on a server equipped with an Intel Xeon Gold 6258R processor (2.70 GHz), 503 GB of RAM, and an NVIDIA A100 GPU (80 GB). The system ran Ubuntu 18.04.5 LTS, with Python 3.7.11, NumPy 1.21.2, and PyTorch 1.10.2. For the experiments in the Enhancement of KV Cache Optimization part, we used datasets from the LongBench tasks, including NarrativeQA, Qaspe, MultiFieldQA-en, MultiFieldQA-zh, HotpotQA, 2WikiMultihopQA, MuSiQue, and DuReader.

For fair comparison and reproducibility, we align the hyperparameter configurations of all baseline and enhancement methods evaluated in our experiments. Specifically:

- For our proposed method, VAM, we use $\alpha = 0.35$ by default in all experiments reported in Table 2, based on empirical tuning that balances contextual integration and token-level fidelity.
- PyramidKV, H2O, SnapKV, and ZipCache are configured with a fixed KV cache size of 2048, following their recommended settings for long-context scenarios.

## C    DATASET AND EVALUATION METRIC MAPPING

Table 3 provides the detailed mapping between dataset identifiers (IDs) and their corresponding datasets, as well as the evaluation metrics used in the experiments. This mapping ensures clarity and reproducibility, following the approach of Bai et al. (2024), and complements the results presented in the main text.

---

**Algorithm 1** VAM: Value-Attention Merging for KV Cache Update

---

**Require:** Sequence length $L$, model dimension $d_k$, KV cache $(K_{\text{cache}}, V_{\text{cache}})$, parameter $\alpha$

1: Initialize $K_{\text{cache}} \leftarrow [\,]$, $V_{\text{cache}} \leftarrow [\,]$
2: **Phase 1: Pre-Fill Phase**
3: **Objective:** Populate the KV cache for input tokens.
4: **for** $t = 1$ to $L_{\text{input}}$ **do**
5:     Compute key $K_t$ and value $V_t$ for input token $t$.
6:     Append $K_t$ to the key cache:

$$K_{\text{cache}} \leftarrow [K_{\text{cache}}, K_t]$$

7:     Append $V_t$ to the value cache:

$$V_{\text{cache}} \leftarrow [V_{\text{cache}}, V_t]$$

8: **Phase 2: Decode Phase**
9: **Objective:** Dynamically update the value cache during autoregressive decoding.
10: **for** $t = 1$ to $L_{\text{decode}}$ **do**
11:     **Key Calculation:**
12:     Compute $K_t$ for the current token and append to $K_{\text{cache}}$:

$$K_{\text{cache}} \leftarrow [K_{\text{cache}}, K_t]$$

13:     **Value Calculation:**
14:     Compute $V_t$ for the current token (as in standard transformers).
15:     **Attention Output Calculation:**
16:     Compute the attention output using $Q_t$, $K_{\text{cache}}$, and $V_{\text{cache}}$:

$$\text{Attn\_Output}_t = \text{softmax}\left(\frac{Q_t K_{\text{cache}}^\top}{\sqrt{d_k}}\right) V_{\text{cache}}$$

17:     **Dynamic Value Update:**
18:     Update $V_t$ by merging it with $\text{Attn\_Output}_t$:

$$V_t^{\text{updated}} \leftarrow V_t + \alpha \cdot \text{Attn\_Output}_t$$

19:     **Cache Update:**
20:     Append the updated value to $V_{\text{cache}}$:

$$V_{\text{cache}} \leftarrow [V_{\text{cache}}, V_t^{\text{updated}}]$$

21: **Output:** Updated KV cache $(K_{\text{cache}}, V_{\text{cache}})$

---

# D  DETAIL OF LLMS

We provide the links to the details of the LLMs used in our experiments in Table 4.

# E  T-SNE VISUALIZATION: RESOLVING LONG-TEXT CONTEXTUAL DEGRADATION

To further investigate the effectiveness of our proposed method in preserving semantic consistency over long sequences, we conducted a t-SNE analysis of token embeddings across the inference process. For this experiment, the interpolation coefficient $\alpha$ was fixed at 0.35. The t-SNE plots presented in Figure 7 compare the contextual token representations between the original model and our enhanced version.

Table 3: An overview of the dataset statistics in LongBench. Chinese datasets are highlighted. 'Source' denotes the origin of the context. 'Avg len' (average length) is computed using the number of words for the English (code) datasets and the number of characters for the Chinese datasets. 'Accuracy (CLS)' refers to classification accuracy, while 'Accuracy (EM)' refers to exact match accuracy.

| Dataset | ID | Source | Avg len | Metric | Language | #data |
|---|---|---|---|---|---|---|
| *Single-Document QA* | | | | | | |
| NarrativeQA | 1-1 | Literature, Film | 18,409 | F1 | English | 200 |
| Qasper | 1-2 | Science | 3,619 | F1 | English | 200 |
| MultiFieldQA-en | 1-3 | Multi-field | 4,559 | F1 | English | 150 |
| MultiFieldQA-zh | 1-4 | Multi-field | 6,701 | F1 | Chinese | 200 |
| *Multi-Document QA* | | | | | | |
| HotpotQA | 2-1 | Wikipedia | 9,151 | F1 | English | 200 |
| 2WikiMultihopQA | 2-2 | Wikipedia | 4,887 | F1 | English | 200 |
| MuSiQue | 2-3 | Wikipedia | 11,214 | F1 | English | 200 |
| DuReader | 2-4 | Baidu Search | 15,768 | Rouge-L | Chinese | 200 |
| *Summarization* | | | | | | |
| GovReport | 3-1 | Government report | 8,734 | Rouge-L | English | 200 |
| QMSum | 3-2 | Meeting | 10,614 | Rouge-L | English | 200 |
| MultiNews | 3-3 | News | 2,113 | Rouge-L | English | 200 |
| VCSUM | 3-4 | Meeting | 15,380 | Rouge-L | Chinese | 200 |

Table 4: LLMs used in the experiments

| Model Name | URL |
|---|---|
| Llama-2-7b-chat-hf | https://huggingface.co/meta-llama/Llama-2-7b-chat-hf |
| Llama-3.1-8B-Instruct | https://huggingface.co/meta-llama/Llama-3.1-8B-Instruct |
| Llama-3.1-70B-Instruct | https://huggingface.co/meta-llama/Llama-3.1-70B-Instruct |
| Mistral-7B-Instruct-v0.3 | https://huggingface.co/mistralai/Mistral-7B-Instruct-v0.3 |

# F  USE OF LLMS

LLMs were used solely for minor language refinement, including improvements in clarity, grammar, and readability. All conceptual development, technical contributions, experimental design, and results are entirely the responsibility of the authors.

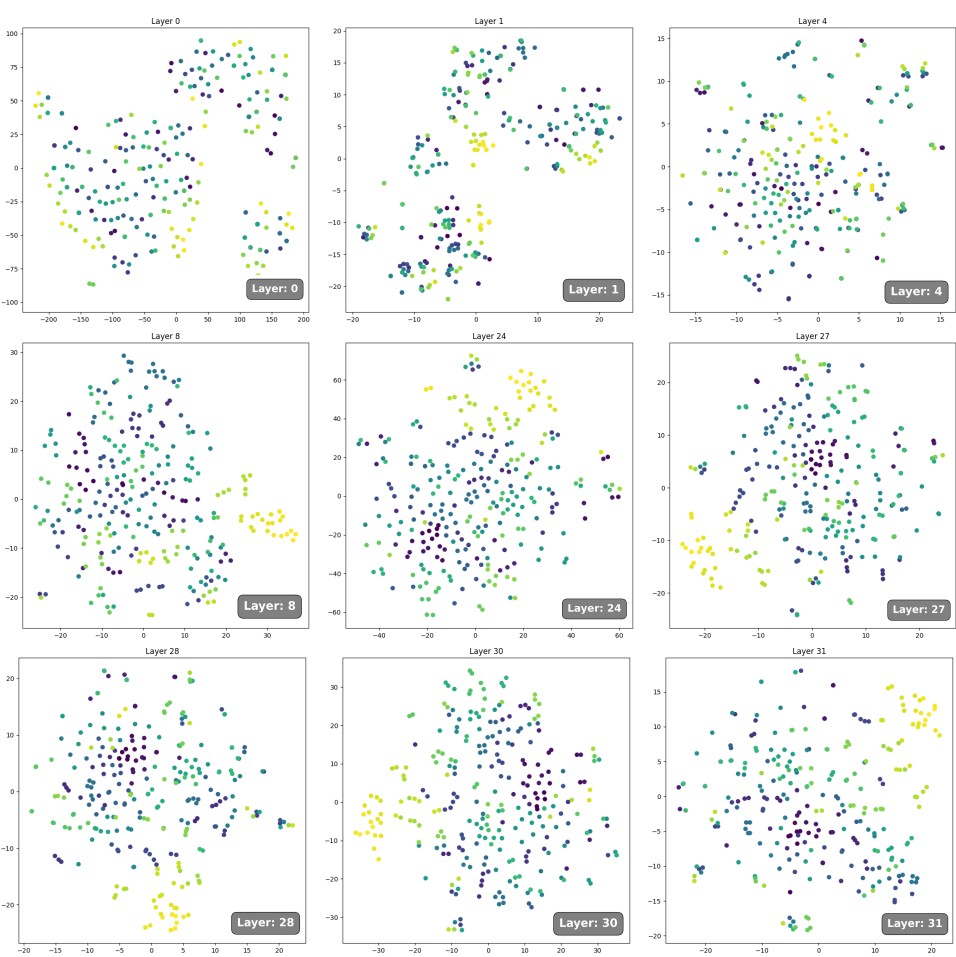

Figure 7: t-SNE visualization of the values stored in the KV cache of the LLaMA2-7B model after applying our method. The visualization shows a more uniform distribution of token values, maintaining a global context focus even in deeper layers, indicating the effective mitigation of context degradation.

