# OpenReview forum: "VAM: Value-Attention Merging for KV Cache Optimization in LLMs"
_ICLR.cc/2026/Conference — ICLR 2026 Conference Withdrawn Submission_

### Official Review · Reviewer_1RsN · 2025-10-20

**Soundness:** 2
**Presentation:** 3
**Contribution:** 2
**Rating:** 2
**Confidence:** 4

**Summary:**

The authors propose a method that optimizes the KV Cache's representation (specifically, its preservation of contextual semantics), which enhances the generation quality of the original model after kv cache compression.

**Strengths:**

- The paper is easy to follow.
- The visualization is clear.

**Weaknesses:**

- The paper lacks sufficient theoretical justification and robust experimental validation.
- The observed performance gains are marginal and appear unstable across different settings.
- The visualization analysis is based on an overly old model (Llama2-7B). Directly applying the conclusions drawn from this limited analysis to other models severely compromises confidence.
- The core idea—directly applying reweighting to the Value Cache before storage—is overly simplistic and lacks a formal theoretical analysis. Furthermore, similar methodologies have been previously proposed in existing work, notably CAM[1].
- The experiments on the LongBench benchmark are missing results for the complete set of datasets (a full 16 datasets are typically expected to against different base methods).
- The proposed "Progressive Clustering" observation appears to be an inherent property of model generation, analogous to the principle that a more detailed or better-crafted prompt leads to more deterministic output.

References:

CaM: Cache Merging for Memory-efficient LLMs Inference. ICML 2024

**Questions:**

- The proposed method is similar to existing work, CAM. Detailed explanation of the differences and compare is required.
- Please supplement the manuscript with a complete set of results for all 16 datasets in the LongBench.

---

### Official Review · Reviewer_PNdu · 2025-10-25

**Soundness:** 3
**Presentation:** 3
**Contribution:** 2
**Rating:** 4
**Confidence:** 3

**Summary:**

This paper introduces a method to integrate contextual information into the stored value states at inference time. Experiments on a variety of tasks show that this improves performance slightly. The simplicity of this method makes it easy to test and deploy, though I think the benefit is a bit incremental. I think it may be worthwhile to explore more deeply as to why the value states seem fairly robust to this perturbation.

**Strengths:**

1. Simple method which does not need any training.
2. Consistent improvement across models and tasks.
3. Visualizations of current limitations are very helpful.

**Weaknesses:**

1. While consistent, the improvement that VAM adds on top of LLMs is fairly marginal. There seems to be a slightly larger benefit when used in conjunction with sparse attention methods, which I think may warrant further investigation.
2. How does VAM perform on long generation tasks like CoT math?
3. Table 2: I think it would be beneficial to include a row for the full KV cache for comparison.
4. See questions

**Questions:**

1. How does this work with multiquery/group query attention? Wouldn't a separate value state need to be stored for each query head, negating half of the memory benefit of these methods?
2. Have you tried fine tuning LLMs with VAM? I wonder if this can help with the training process.
3. How does VAM affect the output token distribution? I'm wondering if the logits become more/less concentrated.

---

### Official Review · Reviewer_jVFV · 2025-10-31

**Soundness:** 2
**Presentation:** 3
**Contribution:** 2
**Rating:** 2
**Confidence:** 4

**Summary:**

This paper proposes VAM (Value-Attention Merging), a plug-and-play algorithm for optimizing the KV cache in LLMs during long-text inference. The authors identify a limitation of standard KV caching: Context Degradation (token representations become increasingly localized) and Context Degradation (as the sequence lenth grows, the model increasingly prioritizes recent tokens). To mitigate these issues, VAM dynamically updates the value cache by merging the original cache vector with the attention output of the first layer, offering more contextual information into the cache.
Experiment on the LongBench benchmark, showing consistent performance improvements  on LLaMA and Mistral models.

**Strengths:**

- The paper addresses a critical and problem in long-context LLM inference (missing contextual information under long-context settings).
- The method is training free and easy to implement
- VAM is a plug-and-play method, can be combined with other KV cache acceleration methods to further improve their performance.

**Weaknesses:**

- The experiments are unconvincing: the models used are outdated, with most experiments on older models like LLaMA-2-7B and Mistral-7B, whose context windows are much smaller than models used in other KV cache papers (e.g., LLaMA-3.1-8B-1024K, Phi-3-Mini-128K...). Secondly, the RULER benchmark should be tested, since it is a more accepted dataset for long-context scenarios with context lengths up to 256k.
- Motivation: The author use t-SNE visualization on the LLaMA2-7B model to demonstrate the Context Degradation phenomenon, which is not convincing. The authors should verify that this phenomenon is widespread on a variety of different model architecture, especially those with a large context window.
- Overhead Analysis: The absence of concrete latency or memory overhead measurements is a notable omission. Claims of "negligible overhead" must be substantiated with statistics (e.g., tps, latency).

**Questions:**

- There is a typo in the caption of Figure 4, the method is VAM, not VAMP?
- For Table2, instead of only 5 tasks, why not report results on other tasks of LongBench?

---

### Official Review · Reviewer_8frj · 2025-11-01

**Soundness:** 2
**Presentation:** 3
**Contribution:** 3
**Rating:** 4
**Confidence:** 4

**Summary:**

The paper is interesting in the way it tries to make KV cache not just smaller but smarter, the motivation makes sense, but the design is still very heuristic.

The proposed method wants to make value representations more contextual instead of static.
The idea to merge the current attention output into each value vector is simple and appealing, but it also assumes that the two vectors live in the same neural space, which is not strictly true, so the addition is more of a practical shortcut than a well grounded mathematical operation.

This work shows good results on LongBench but doesn’t test on reasoning heavy or extreme length sets like RULER or InfiniteBench, so the claim of general long context benefit feels limited. also, all tests are on llama and mistral, which share the same cache interface. it doesn’t include Qwen or DeepSeek, whose fused attention kernels would make this merging step non-trivial.

The merge behaves like a residual update or an exponential moving average, letting each cached value carry some global semantic bias. it improves coherence within single long sequences, but if applied blindly across different modes of reasoning or tool use, it could blur boundaries between functions or roles. In an agentic setting with function calls or json interactions, this could easily mix unrelated context and weaken state tracking.

The method is plug-and-play and can combine with any compression method, however,  the weak part is the lack of deeper explanation, limited coverage, and a strong reliance on linear addition that may not hold under architectural diversity.

Overall, it’s a empirical approach, light and effective for classic long-context tasks, but it probably needs a more structured or gated variant to stay useful in agentic LLMs.

**Strengths:**

1. The method is simple, plug-and-play, and doesn’t need retraining.
2. It improves cache quality with almost no cost and works well with other KV compression approaches.
3. Results on LongBench are consistent, and visual evidence supports its effect.

**Weaknesses:**

1. It’s limited in scope and theory. the method is only tested on llama and mistral
2. The addition of attention output to values assumes both lie in the same neural space, which isn’t strictly true.
3. The fixed α makes it inflexible, and over merging can blur semantics.
4. In agentic or multi-modal scenarios, it could mix unrelated context and break function separation.

**Questions:**

1. How exactly does the linear merge avoid subspace misalignment between value and attention outputs?
2. Would a projection or learned adapter make it more stable across architectures?
3. Why only test on llama and mistral—would models with fused attention kernels behave differently?
4. Does the merge still help when the context exceeds LongBench’s length?
5. How would the method perform in agentic settings?
6. Is the semantic benefit still consistent when tokens switch roles, like between reasoning and tool calling?

---

### Note · Authors · 2025-12-05

I have read and agree with the venue's withdrawal policy on behalf of myself and my co-authors.